# MapSelect: Sparse & Interpretable Graph Attention Networks

## Abstract

Graph Attention Networks (GATs) have shown remarkable performance in capturing complex graph structures by assigning dense attention weights over all neighbours of a node. Attention weights can act as an inherent explanation for the model output, by highlighting the most important neighbours for a given input graph. However, the dense nature of the attention layer causes a lack of focus as all edges receive some probability mass. To overcome this, we introduce *MapSelect*, a new method providing a fully differentiable sparse attention mechanism. Through user-defined constraints, MapSelect enables precise control over the attention density, acting as a continuous relaxation of the popular top-$k$ operator. We propose two distinct variants of MapSelect: a local approach maintaining a fixed degree per node, and a global approach preserving a percentage of the full graph. Upon conducting a comprehensive evaluation of five sparse GATs in terms of sparsity, performance, and interpretability, we provide insights on the sparsity-accuracy and sparsity-interpretability trade-offs. Our results show that MapSelect outperforms robust baselines in terms of interpretability, especially in the local context, while also leading to competitive task performance on real-world datasets.

## 1 Introduction

Graph Attention Networks (GATs) employ the attention mechanism to weigh the importance of neighbours of a node and their features when aggregating information. This ultimately allows for a more adaptive learning for the task at hand (Veličković et al., 2018). The learned attention weights can also be inspected to gain insights into what the model considers as discriminative features towards a final decision (Ying et al., 2019; Ye & Ji, 2021; Rath et al., 2021). However, the dense nature of the attention mechanism caused by the softmax transformation, which assigns probability mass to all edges (even irrelevant ones) challenges interpretability, thereby resulting in a computation graph as dense as the input graph itself. Thresholding attention probabilities is a straightforward solution to the issue of dense attention, but it compromises the end-to-end differentiability of the network.

The importance of sparsifying GATs has been recognized by a number of works with applications to robustness, task performance, and computational efficiency (Kipf et al., 2018; Srinivasa et al., 2020; Luo et al., 2021; Shirzad et al., 2023; Ye & Ji, 2021). Particularly, the work by Rathee et al. (2021) rely on sparsity to improve interpretability. Such methods are known as self-interpretable approaches, where explanations are an integral part of the model's decision process. Consequently, self-interpretable techniques are renowned for producing more faithful explanations (Jacovi & Goldberg, 2020a; Wiegreffe & Pinter, 2019). However, it is important to note that these approaches typically involve a trade-off between task performance and interpretability, and the effects of sparsity-inducing hyperparameters on this trade-off remain unclear Moreover, even if there is a focus on interpretability, the ability to control the induced subgraph size, and thereby the interpretability, is often sidestepped (Rathee et al., 2021).

In this paper, motivated by the success of controllable sparse attention methods in NLP (Correia et al., 2019; Treviso & Martins, 2020; Guerreiro & Martins, 2021), we focus on self-interpretable methods for GNNs, and develop a novel framework named *MapSelect* that produces sparse controllable subgraphs while maintaining a high task accuracy. In particular, we use SparseMAP (Niculae et al., 2018) to create *differentiable* sparse attention masks. Differently from alternative solutions, the proposed framework can be controlled both locally and globally, through two configurations:

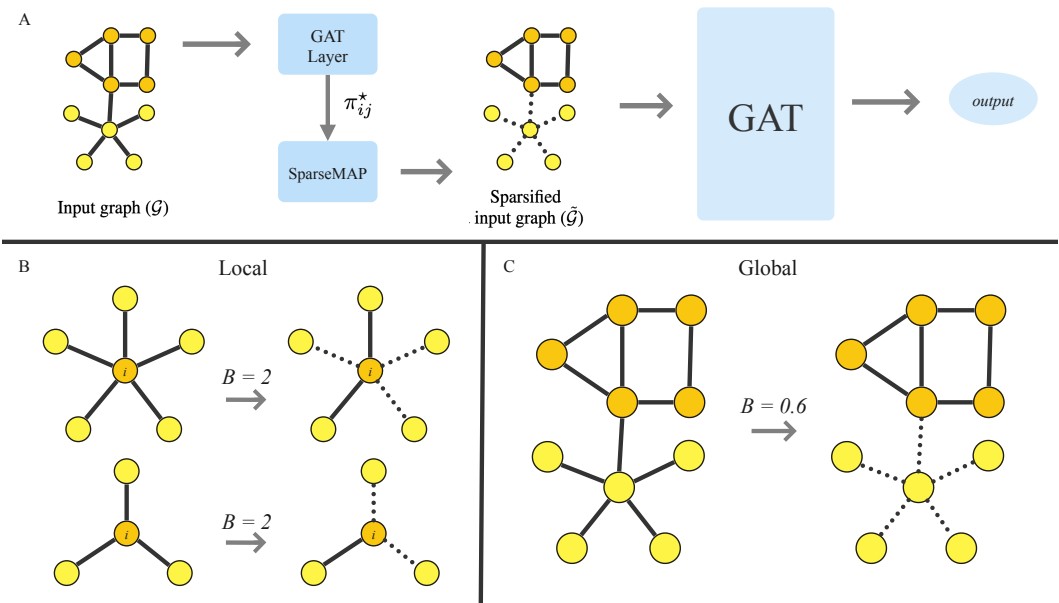

Figure 1: Overview of MapSelect. (A) The input graph is sparsified by applying SparseMAP (see §2.2) in a local or global fashion, conditioned to the information processed by a GAT layer. (B) In the local approach, MapSelect removes edges within the neighbourhood of a node, with $B$ representing the maximum number of active connections. (C) In the global approach, MapSelect sparsifies the full graph, with $B$ denoting the portion of active edges. In this example, in order to identify a "house" structure within a graph, MapSelect-G retains only the edges in the "house" structure.

(i) *MapSelect-L*, which produces an attention mask that maintains only the essential edges per node based on a fixed budget and (ii) *MapSelect-G*, a configuration that only maintains the most essential edges in the full graph based on a target budget. Both configurations allow for an easy control to capture the most essential edges that will provide a more focused attention mask, allowing to identify important substructures, as illustrated in Figure 1.

Using five node-level benchmark datasets, we study the effect of sparsity on both performance and interpretability, ultimately, establishing a trade-off that provides deeper insights into interpretability. We compare our method against five baselines and we also validate on a dataset with a ground-truth explanation to highlight the explanatory capability of the learned sparse attention weights. We find that MapSelect presents itself as the only method to consistently improve interpretability across all datasets, and especially on denser graphs. Overall, our contribution is twofold:[1]

1. We propose *MapSelect* to control the sparsity of graph attention layers, both locally and globally, leading to superior interpretability results compared to baselines.

2. We provide an extensive and unique evaluation of sparse GATs, examining trade-offs between sparsity-accuracy and sparsity-interpretability, and providing insights into architectural choices that influence interpretability.

## 2 BACKGROUND

We denote a directed graph as $\mathcal{G} = (\mathcal{V}, \mathcal{E})$ with nodes $\mathcal{V} = \{1, ..., N\}$ and edges $\mathcal{E} \subseteq \mathcal{V} \times \mathcal{V}$, where $(j, i) \in \mathcal{E}$ represents an edge from $j$ to $i$, and $\mathcal{N}_i = \{j \in \mathcal{V} \mid (j, i) \in \mathcal{E}\}$ the neighborhood of node $i$.

---

[1]Our code is available at: `blind review.`

## 2.1 GRAPH ATTENTION NETWORKS

A GAT layer computes a weighted average of vector representation of the neighbors of a node (Veličković et al., 2018). Specifically, given a set of node representations $\{\boldsymbol{h}_i^{(0)} \in \mathbb{R}^d \mid i \in \mathcal{V}\}$ as input (at layer $\ell = 0$), a GAT layer first computes attention scores for edges $(i, j) \in \mathcal{E}$ as:[2]

$$z_{ij}^{(\ell)} = \boldsymbol{a}^{(\ell)} \cdot \mathrm{LeakyReLU}\left(\mathrm{concat}(\boldsymbol{W}_1^{(\ell)} \boldsymbol{h}_i^{(\ell)}, \boldsymbol{W}_2^{(\ell)} \boldsymbol{h}_j^{(\ell)})\right), \tag{1}$$

where $\boldsymbol{a}^{(\ell)} \in \mathbb{R}^{d'}$, $\boldsymbol{W}_1^{(\ell)}, \boldsymbol{W}_2^{(\ell)} \in \mathbb{R}^{d' \times d}$ are learnable linear transformations, and $\mathrm{concat}(\cdot)$ denotes vector concatenation. Attention weights are then obtained by employing the softmax transformation:

$$\pi_{ij}^{(\ell)} = \mathrm{softmax}(\boldsymbol{z}_i^{(\ell)})_j := \frac{\exp(z_{ij}^{(\ell)})}{\sum_{j' \in \mathcal{N}_i} \exp(z_{ij'}^{(\ell)})}, \tag{2}$$

where $\boldsymbol{z}_i^{(\ell)} \in \mathbb{R}^n$ represents the attention scores of node $i$, with $n = |\mathcal{N}_i|$. That is, softmax maps scores to probabilities $\mathbb{R}^n \to \triangle_{n-1}$, where $\triangle_{n-1} := \{\boldsymbol{\xi} \in \mathbb{R}^n \mid \boldsymbol{\xi} \geq \boldsymbol{0},\ \boldsymbol{1}^\top \boldsymbol{\xi} = 1\}$ is the $(n-1)$-probability simplex. The updated representation of node $i$, $\boldsymbol{h}_i^{(\ell+1)} \in \mathbb{R}^{d'}$, is determined by the weighted average of the transformed features from neighbouring nodes, potentially followed by a non-linear function $\sigma(\cdot)$:

$$\boldsymbol{h}_i^{(\ell+1)} = \sigma\left(\sum_{j \in \mathcal{N}_i} \pi_{ij}^{(\ell)} \boldsymbol{W}_2^{(\ell)} \boldsymbol{h}_j^{(\ell)}\right). \tag{3}$$

While softmax is easy to implement and fully differentiable, its output is dense and thus all edges will have some probability mass, which may hinder interpretability. This has been investigated in (Treviso & Martins, 2020) for NLP tasks.

## 2.2 SPARSE ATTENTION

Previous studies have shown that incorporating sparsity into the attention mechanism results in a more compact and transparent representation of $\mathcal{G}$ (Kipf et al., 2018; Ye & Ji, 2021). In this section, we briefly review SparseMAP (Niculae et al., 2018), a technique that achieves this functionality while preserving the necessary differentiability for backpropagation. In §3, we leverage SparseMAP to introduce a new sparse attention method for GNNs.

Let $\boldsymbol{z} \in \mathbb{R}^n$ be a vector of scores given to the edges of a particular node. To improve interpretability, a possible approach is to transform $\boldsymbol{z}$ into a *sparse* probability vector $\boldsymbol{\pi}$ whose entries indicate in probability the role of an edge towards the final decision. This can be achieved by the $\alpha$-entmax attention (Peters et al., 2019), a generalization of softmax, which has been to obtain sparse transformers (Correia et al., 2019). For situations where one wants to specify precisely the number of neighbors, a top-$k$ operation can also be used, such that only the $k$ largest entries in $\boldsymbol{\pi}$ are kept, while the others are zeroed out (Gao & Ji, 2019). However, the top-$k$ operation is not differentiable and may inevitably introduce instabilities for training .

SparseMAP addresses this issue by casting subset selection as a relaxed structured prediction problem. Specifically, when considering $n$-length binary sequences with at most $B$ non-zeros, SparseMAP is defined as:

$$\mathrm{SparseMAP}(\boldsymbol{z}; B) := \underset{\boldsymbol{\mu} \in \mathcal{M}_\mathcal{S}}{\arg\max}\ \boldsymbol{z}^\top \boldsymbol{\mu} + \frac{1}{2}\|\boldsymbol{\mu}\|_2^2$$

$$\mathrm{s.t.}\quad \mathcal{M}_\mathcal{S} := \Big\{\sum_i \pi_i \boldsymbol{y}_i : \boldsymbol{\pi} \in \triangle_{|\mathcal{S}|-1}, \boldsymbol{y}_i \in \mathcal{S}\Big\} \tag{4}$$

$$\mathcal{S} := \{\boldsymbol{y} \in \{0,1\}^n \mid \|\boldsymbol{y}\|_1 \leq B\},$$

where the cost function aims at finding a vector $\boldsymbol{\mu}$ that is aligned with the scores $\boldsymbol{z}$ but with a quadratic regularizer to ensure smoothness. The solution is confined to the marginal polytope $\mathcal{M}_\mathcal{S}$

---

[2]We adopt the GAT variant proposed by (Brody et al., 2022), called GATv2, due to its superior expressivity.

that imposes solutions to be in the space of bit vectors $[0, 1]^n$ with at most $B$ non-zeros. Notably, the vertices of $\mathcal{M}_S$ represent binary solutions and in such cases we obtain $u \in \{0, 1\}^n$, whereas its edges represent a sparse convex combination of binary vectors, and thus $u \in [0, 1]^n$. Finally, the faces of this polytope lead to fully dense solutions. Because of the quadratic regularization term, SparseMAP promotes sparse vectors $u \in [0, 1]^n$ that lie on the boundary of the marginal polytope (vertices, edges, or other low-dimensional faces). For more information on SparseMAP, we refer the reader to (Niculae et al., 2018; Niculae & Martins, 2020).

Therefore, given the vector scores $z_i \in \mathbb{R}^n$ of node $i \in \mathcal{V}$, SparseMAP will produce a vector $\mu_i \in [0, 1]^n$ as output, such that edges with $\mu_{ij} = 0$ can be ignored during the forward pass. While the optimization problem described in Equation 4 does not have a closed-form solution, both the forward and backward passes can be solved with an active set method that exhibits exact finite convergence and yields the optimal sparsity pattern (Nocedal & Wright, 1999). Contrarily to stochastic approaches, such as the reparameterization trick used in NeuralSparse (Zheng et al., 2020) and SGAT (Ye & Ji, 2021), SparseMAP is deterministic and end-to-end differentiable, and thus easier to optimize (Guerreiro & Martins, 2021). Next, we present MapSelect, a new sparse method for GNNs that leverages SparseMAP.

## 3   MAPSELECT

We introduce two methods that leverage sparsity to design more interpretable GNNs by acting on different levels of the computation graph. The first method, **MapSelect-L**, keeps a sparse subset of local connections for each node, while the second approach, **MapSelect-G**, promotes sparsity on the full computation graph; see Figure 1 for an overview. For both approaches, we start with a GAT layer that takes the input graph representation and produces the attention scores $z_{ij}^\star$ and the attention weights $\pi_{ij}^\star$ for each edge $(i, j) \in \mathcal{E}$, as described in §2.1. EWe pass the attention weights (MapSelect-L) or the attention scores (MapSelect-G) to SparseMAP (cf. Equation 4) and obtain a sparse distribution as output, which we leverage to obtain a sparse input graph $\tilde{\mathcal{G}}$ that is used in subsequent GAT layers. Next, we detail each variant of MapSelect.

**MapSelect-L.**   In this approach, we fix a local budget $B$ per node, such that each node keeps at most $B$ active connections. Formally, for each node $i$, let $\pi_i^\star \in \triangle_{n-1}$ be the attention weights obtained with the first GAT layer, where $n = |\mathcal{N}(i)|$ and let $\text{SparseMAP}(\cdot; B)$ denote the SparseMAP with a budget constraint $B$. In MapSelect-L, we apply SparseMAP on $\pi_i^\star$ with a budget constraint to get a sparse mask $\mu_i \in [0, 1]^n$:

$$\mu_i = \text{SparseMAP}(\pi_i^\star / t; B), \tag{5}$$

where $t \in \mathbb{R}$ is a temperature hyperparameter. Next, we use $\mu_i$ to re-scale the attention weights of each subsequent $\ell$-th GAT layer as:[3]

$$\tilde{\pi}_i^{(\ell)} = \frac{\mu_i \odot \pi_i^{(\ell)}}{\mu_i^\top \pi_i^{(\ell)}}, \quad h_i^{(\ell+1)} = \sigma\left(\sum_{j \in \mathcal{N}_i} \tilde{\pi}_{ij}^{(\ell)} W_2^{(\ell)} h_j^{(\ell)}\right), \tag{6}$$

where $\odot$ represents the element-wise multiplication, and $h_i^{(0)}$ the feature vector fed to the network (see Figure 1). This procedure effectively deactivates the contribution of neighbouring nodes $j \in \mathcal{N}_i$ when $\mu_{ij} = 0$. In other words, we condition SparseMAP on information processed by a GAT layer and then use its output to sparsify the input graph by adjusting subsequent attention layers. Notably, as the temperature $t \to 0$, SparseMAP becomes the top-$B$ operator and $\tilde{\pi}_i$ becomes a re-normalized vector of probabilities with the top-$B$ highest original probabilities. Therefore, our proposed framework can also be seen as a continuous relaxation of the usual truncation approach.[4]

**Remark 1.** The current approach imposes the budget $B$ as the maximum number of edges allowed per node. In some cases, where statistical properties of the input graph (such as degree distribution or centrality metrics) may be relevant this strategy can be changed by using a relative budget ($B\%$) on the available number of edges. In our experiments, we tested both approaches but have not noticed a significant impact on performance.   ∎

---

[3]We ensure $\mu_i \neq 0$ by keeping self-loops (i.e., $\mu_{ii} = 1$ always).

[4]We employ a temperature parameter of $10^{-1}$ and $10^{-3}$ in training and test time, respectively.

**MapSelect-G.** MapSelect-G evokes SparseMAP over all edges globally, disregarding specific neighbourhoods. Here, we set the budget $B$ as a percentage of the number of edges as:

$$\bar{\boldsymbol{z}}^{\star} = \text{concat-and-pad}(\boldsymbol{z}_1^{\star}, ..., \boldsymbol{z}_N^{\star}) \tag{7}$$

$$\bar{\boldsymbol{\mu}} = \text{SparseMAP}(\bar{\boldsymbol{z}}^{\star}/t; B), \tag{8}$$

where $\bar{\boldsymbol{z}}^{\star} \in \mathbb{R}^{N^2}$ is the concatenation of all attention scores $\boldsymbol{z}_i^{\star}$ from the 1st GAT layer for all $1 \leq i \leq N$ nodes, padding $(i, j)$ positions with $-\infty$ when $(i, j) \notin \mathcal{E}$. Here, we denote $\boldsymbol{\mu}_i \in [0, 1]^N$ as the binary vector given to node $i$, indexed as the $i$-th contiguous chunk of size $N$ in $\bar{\boldsymbol{\mu}} \in [0, 1]^{N^2}$. As in MapSelect-L, we use $\boldsymbol{\mu}_i$ to re-scale the attention weights of node $i$ in subsequent GAT layers (see Equation 6), keeping self-loops by setting $\mu_{ii} = 1$. Therefore, edges $(i, j) \in \mathcal{E}$ will be deactivated whenever $\mu_{ij} = 0$, and as a result, they will not contribute towards the final output.

**Connections with related approaches.** Both MapSelect-L and MapSelect-G resemble techniques that sparsify the input graph and then use it in a classification task, such as NeuralSparse (Zheng et al., 2020), SGAT (Ye & Ji, 2021), and DropEdge (Rong et al., 2020). However, MapSelect differs by: (i) leveraging SparseMAP to sparsify the input graph, effectively keeping the classification problem end-to-end-differentiable; and (ii) applying the resulting mask to the attention mechanism in subsequent GAT layers instead of masking irrelevant connections directly in the adjacency matrix. More specifically, MapSelect-L is similar to NeuralSparse and the traditional top-$k$ attention, as the selection of relevant edges occurs in the neighbourhood of each node in the computation graph and the selection budget is set to a pre-defined fixed number of edges. MapSelect-G is close in spirit to SGAT and DropEdge, as the decision to deactivate irrelevant edges is carried globally over the entire input graph. Finally, the way MapSelect conditions on the initial GAT layer mirrors the design seen in models termed "rationalizers" within the NLP literature (Lei et al., 2016; Bastings et al., 2019; Guerreiro & Martins, 2021). Much like MapSelect, these models aim to provide faithful explanations by conditioning the selection of input elements (e.g., words) on an encoder module, and subsequently making a final decision solely on the basis of these selected items.

# 4 EVALUATION

We compare the proposed methods to five baselines that focus on producing a sparse subset of the input graph. We perform experiments on five real-world datasets, and on one synthetic dataset containing ground truth explanations. The detailed model configurations and the dataset information can be found in §B.1 and §B.3, respectively.

## 4.1 BASELINES

We assess the proposed approaches by comparing them to the following alternatives. A summary of the characteristics of each method is presented in Table 1.

**Top-$k$.** We apply a top-$k$ operation on the softmax attention weights of a standard two-layer GAT in a local fashion. We control for sparsity by varying $k$ at test time.

**Entmax.** This approach replaces the standard softmax function found in GATs by the $\alpha$-entmax transformation (Peters et al., 2019), detailed in §A. We control the propensity to sparsity by varying $\alpha$. Both Top-$k$ and Entmax produce sparse attention probabilities directly rather than implementing a separate attention layer that masks the input graph.

Table 1: Characteristics of each baseline method.

| Method | Sparsity Level | Sparsity Control | End-to-end Differentiable |
|---|---|---|---|
| Top-$k$ | Local | ✓ | ✗ |
| Entmax | Local | ✗ | ✓ |
| NeuralSparse | Local | ✓ | ✗ |
| MapSelect-L | Local | ✓ | ✓ |
| SGAT | Global | ✗ | ✗ |
| DropEdge | Global | ✓ | ✗ |
| MapSelect-G | Global | ✓ | ✓ |

**NeuralSparse. (Zheng et al., 2020)** NeuralSparse utilizes Gumbel-Softmax (Jang et al., 2017) to sample (local) sparse subgraphs consisting of $k$ neighbours. We control sparsity by setting $k$ as the maximum number of edges per node.

**SGAT. (Ye & Ji, 2021)** SGAT encourages sparse solutions by adding an $\ell_0$-penalty term to the loss function, penalizing non-zero attention weights, resulting in global sparsification. SGAT resorts to

the hard concrete estimator for model optimization (Louizos et al., 2018). We control sparsity by adjusting the weight given to the $\ell_0$ penalty empirically.

**DropEdge. (Rong et al., 2020)** DropEdge randomly drops edges from the input graph, thus acting in a global fashion. We consider this method as a baseline by controlling the portion of dropped edges and maintaining the sparsified graph at test time.

## 4.2 EXPERIMENTAL SETUP

**MapSelect.** For both variants of MapSelect, a single GAT layer is employed to derive the set of attention weights $\boldsymbol{\pi}_i^\star$, which are used to form the sparse mask. Following this, two GAT layers are employed to classify the input using the masked attention.[5] We control the sparsity of MapSelect by adjusting the SparseMAP's budget constraint $B$. In the local approach, $B$ can alternatively be configured to retain a specific percentage of connections. However, for the sake of uniformity with NeuralSparse and due to similar performance, we only assess the fixed configuration.

**Metrics.** We evaluate interpretability with the fidelity metric proposed by ZORRO (Funke et al., 2023). More details on fidelity can be found in §B.2. Since the synthetic dataset provides binary vectors as ground truth explanations, for this dataset we also compare our explanations with respect to the ground truth in terms of AUC, which automatically accounts for multiple binarization thresholds. Furthermore, an evaluation of the explanation entropy, as proposed by BAGEL (Rathee et al., 2022), is presented in §D.1.

**Explanation extraction.** We employ two distinct strategies for extracting explanations. For real-world datasets, we obtain *node-level explanations* by propagating an identity matrix over the computation graph. We detail this strategy in §C. For the synthetic dataset, we follow the approach proposed by (Ying et al., 2019), which produces *edge-level explanations* by averaging the attention scores of all layers.

## 4.3 EXPERIMENTS

We hypothesize that as we progressively remove edges from the computation graph, the performance will decline. In addition, we anticipate that a classification based on fewer edges will be more interpretable To investigate these effects independently and identify a balance between them, we pose the following research questions:

RQ1. *What is the role of sparsity on model performance?*

RQ2. *What is the role of sparsification on model interpretability?*

RQ3. *What is the interpretability-performance trade-off?*

### 4.3.1 ROLE OF SPARSITY ON TASK PERFORMANCE

In Figure 2, we present results for all methods on real-world datasets, with graphs sorted from the least to the most dense.[6] Among the local methods, MapSelect-L consistently outperforms NeuralSparse and top-$k$. Notably, unlike its counterparts, MapSelect-L is not tied to a specified budget, giving us the flexibility to select fewer edges than the targeted allocation. MapSelect-L shows also a more stable convergence as more edges are discarded than the other local approaches.

Looking at the global methods, both MapSelect-G and SGAT surpass DropEdge. In the less dense graphs, SGAT and MapSelect-G achieve similar results, while in more dense datasets SGAT outperforms MapSelect-G. However, SGAT faces challenges in maintaining sparsity control in dense graphs, as it quickly deviates from $10\%$ to $60\%$ sparsity. In contrast, MapSelect-G provides a tight control over sparsity, respecting the desired budget pre-established before training. We provide detailed view of sparsity controllability in §D.3. Regarding the impact of sparsity on accuracy, we note that both SGAT and MapSelect-G can maintain or even improve accuracy as sparsity increases on the Actor dataset, suggesting that this dataset might contain a considerable number of irrelevant

---

[5]To ensure a fair comparison, we use two GAT layers for classification in all methods, including MapSelect, regardless of whether the input is a full or sparsified graph.

[6]Results for a 2-layer dense GAT is recovered by Entmax with $0\%$ edges removed, corresponding to $\alpha = 1$.

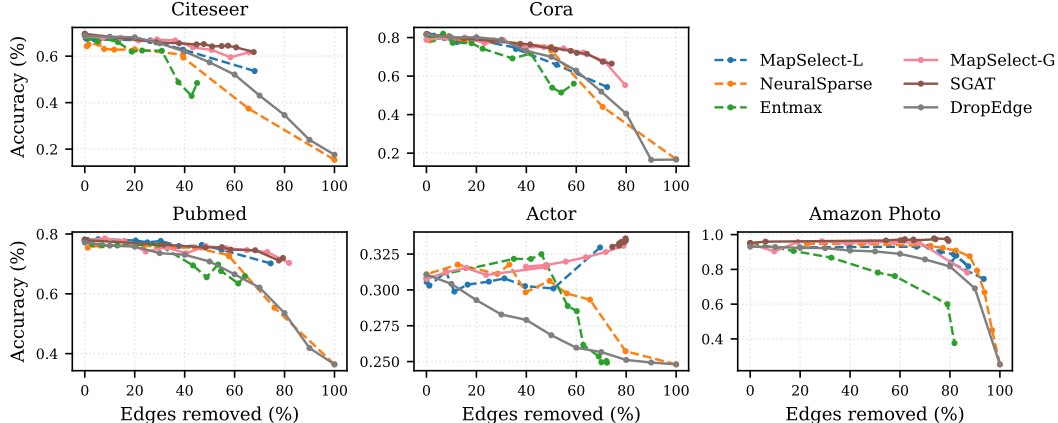

Figure 2: The impact of sparsity on the model performance.

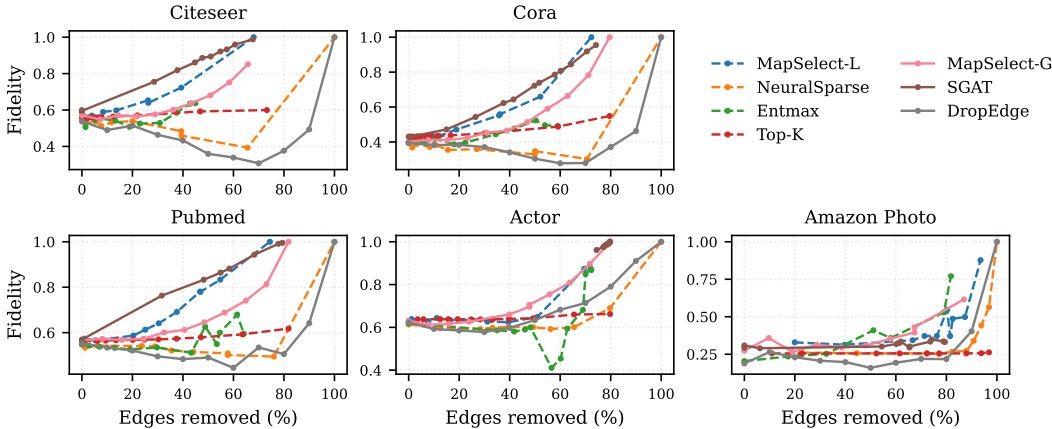

Figure 3: Impact of sparsity on interpretability.

edges. This is expected because global approaches impose fewer limitations on which edges to remove while during training.

Overall, we observe that global methods typically outperform local approaches when the primary focus is on task performance, likely because global approaches impose fewer limitations on edge removal. In a case where it is more beneficial to maintain all edges for one node and remove all edges for a different one, a global approach should be considered. Interestingly though, the local approaches achieve a similar performance on the more dense datasets and sometimes they outperform the global approaches in the most sparsified settings. In addressing RQ1, we find that sparsity presents a nuanced trade-off in task performance. While extreme sparsity can indeed lead to decreased performance, a moderate degree of sparsity, around 40%, results in a minimal performance drop, often less than 5% across all datasets, especially on denser ones.

### 4.3.2 ROLE OF SPARSITY ON INTERPRETABILITY

Towards answering RQ2, we evaluate the tradeoff between sparsity and interpretability on real-world datasets first, and then move to the synthetic dataset with ground-truth explanations.

**Real-world datasets.** In Figure 3, we present the impact of sparsity on fidelity. Intuitively, a high fidelity implies that the explanation is more faithful and is more robust to perturbations. Among the local methods, the results vary as we increase the sparsity rate. Initially, top-$k$ has a better fidelity than MapSelect-L, however, as we remove more edges, MapSelect-L consistently outperforms other methods. The early success of Entmax and top-$k$ might be due to a better attention distribution. This is supported by the lower entropy of Entmax, explored in §D.1. Regarding the global approaches, MapSelect-G outperforms DropEdge and SGAT is the best performing among the global methods.

In contrast to our findings in terms of the sparsity-accuracy tradeoff, global methods do not always lead to a better interpretability than local the approaches. In the context of MapSelect, we see a trend towards preferring local sparsification, which achieves a results competitive to SGAT. Lastly, we remark that graph density significantly impacts the interpretability. For example, while SGAT has the overall best interpretability results, MapSelect-L outperforms it in the Amazon Photo dataset.

**Synthetic dataset.**  To investigate whether an extracted rationale agrees with the ground truth explanation, we evaluate the models on the BA-Shapes dataset. We show the trade-off between sparsity and AUC in Figure 4. The task accuracy of each method can be found in §D.2.

For local methods, we observe a standout performance from MapSelect-L. As more edges are removed, its AUC score increases. However, after removing more than 60% of edges, the score drops. This indicates that MapSelect-L provides better explanations with a moderate sparsity rate. NeuralSparse shows a more significant increase in AUC but starts with a much lower initial score. Top-$k$ performs as expected, producing less faithful explanations as more edges are removed.

Turning to global methods, SGAT outperforms other approaches, mirroring the trend seen with MapSelect-L. The trajectory of MapSelect-G starts with high AUC scores, and then we get lower scores as sparsity increases. The lower performance of MapSelect-L compared with MapSelect-G can be attributed to the nature of the BA-Shapes dataset, which emphasizes the discovery of small structures within a vast graph, deeming all other edges irrelevant. These irrelevant edges are retained in the local approach, as it keeps only a small absolute number of edges per node.

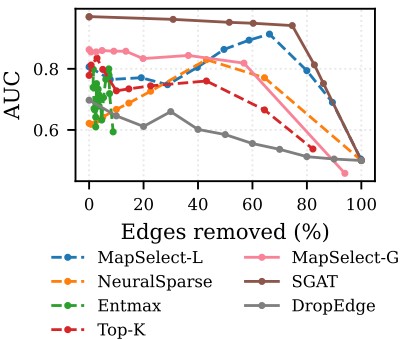

Figure 4: The impact of graph sparsification on the similarity of the extracted rationale to the ground truth explanation (in terms of AUC) on the BA-Shapes dataset.

From the increased trend in the AUC scores, we conclude that attention-based methods can be explored to extract plausible explanations. Notably, both MapSelect-G and SGAT outperform the AUC scores presented by the attention, gradient, and GNNExplainer baselines in (Luo et al., 2020, §5.3). We show an example of an explanation extracted with MapSelect-G for BA-Shapes in Figure 1C.

### 4.3.3 PERFORMANCE-INTERPRETABILITY TRADE-OFF

As seen in Figure 2 and Figure 3, the lowest accuracy and the best fidelity scores are reported when most edges are removed. Since a consistent explanation of a wrong classification may be irrelevant, we investigate the effect of interpretability in accuracy directly in Figure 5. For clarity, we removed the baselines that did not show sufficient improvement in interpretability.

First, we can see that MapSelect-L is the only local method offering an appropriate and consistent trade-off between accuracy and fidelity. Second, we see that MapSelect-G is more suitable for denser datasets (e.g., Amazon Photo), where its ability to control the sparsity allows improving the fidelity score by up to a factor of two while retaining the accuracy. Contrarily, SGAT works best in sparser datasets but in denser ones it strugles to enhance the fidelity due to limited control over edge removal. Both MapSelect methods consistently demonstrate their ability to yield more interpretable networks by robustly removing edges in all scenarios. The preference for either the local or global approach appears to strongly hinge on the task as well as the dataset. Overall, these analyses show that studying the trajectory of task accuracy and interpretability score as we change sparsity reveals a more profound understanding of the capabilities of each method.

## 5 RELATED WORK

Graph sparsification can be applied for a variety of goals, such as robustness, mitigating over-smoothing, removing noise, decreasing computation times or improving interpretability. Here, we focus on existing works that apply sparsity to improve interpretability.

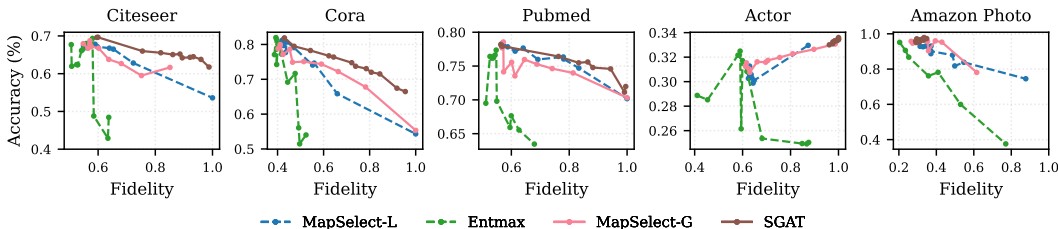

Figure 5: The trade-off between the accuracy and the fidelity score.

**Post-hoc approaches.** The majority of methods targeting the enhancement of GNN interpretability adopt a *post-hoc* approach, pinpointing relevant nodes and features for decisions made by a trained network. Significant contributions in this category include GNNExplainer (Ying et al., 2019), PGExplainer (Luo et al., 2020), XGNN (Yuan et al., 2020), GraphMask (Schlichtkrull et al., 2021), and Zorro (Funke et al., 2023). Although post-hoc methods can be readily applied to any black-box model, they overlook the intrinsic explainable elements of the model, potentially compromising their faithfulness (Rudin, 2019; Kakkad et al., 2023).

**Local self-interpretable methods.** A second perspective of interpretability is given by *self-interpretable* approaches. Contrasting with post-hoc methods, these are integrated directly within the model's architecture. Within this view, the subgraph generated during forward propagation can be considered a faithful explanation for a particular decision. For instance, NeuralSparse (Zheng et al., 2020) learns a $k$-neighbor subgraph for each node by sampling an adjacency from a Gumbel-Softmax distribution and employs the reparametrization trick to address non-differentiability. SEGNN (Meng et al., 2022) constructs an explanation subgraph by grouping $k$ nodes with similar structure and features, learning the grouping process by adding a contrastive penalty to the loss function. In contrast, MapSelect can also act locally within a neighborhood (MapSelect-L), while still being deterministic and end-to-end differentiable without requiring a multi-task objective.

**Global self-interpretable methods.** Other *self-interpretable* methods seek to induce sparsity globally, without restricting this process to a specific neighborhood. This goal is shared by PTD-Net (Luo et al., 2021), KEdge (Rathee et al., 2021), SGAT Ye & Ji (2021), and others (Feng et al., 2022; Zhang et al., 2022; Miao et al., 2022). Analogous to NeuralSparse, PTDNet samples a subgraph that is used for classification, but its sparsity is imposed via a loss penalty, complicating its controllability. KEdge (Rathee et al., 2021) produces a subgraph by sampling binary masks from a HardKuma distribution over the adjacency matrix. Meanwhile, SGAT (Ye & Ji, 2021) prunes edges through attention weights, but requires sampling from a Hard-Concrete distribution. Both KEdge and SGAT resort to the reparameterization trick for optimization. MapSelect-G aligns with SGAT in its methodology, but with SparseMAP, the selection is entirely differentiable and flexible, allowing users to define a specific sparsity budget. Finally, we note that differently to MapSelect, SGAT, PTD-NET, and NeuralSparse do not primarily concentrate on improving interpretability; rather, it emerges as a by-product of their built-in sparse approaches.

**Connections to rationalizers in NLP.** As mentioned in §3, MapSelect aligns with the objectives of rationalizers, colloquially termed *mask-then-predict* techniques, which are prevalent in NLP for extracting faithful explanations (Jain et al., 2020; Jacovi & Goldberg, 2020b). Classical examples include rationalizers that sample masks from a Bernoulli (Lei et al., 2016) or HardKuma distribution (Bastings et al., 2019). Addressing training instabilities triggered by stochastic estimators, Treviso & Martins (2020) suggests leveraging the $\alpha$-entmax transformation (Peters et al., 2019) for the selection mechanism. Guerreiro & Martins (2021) introduced SPECTRA, a method providing differentiability and control over sparsity through SparseMAP (Niculae & Martins, 2020), exhibiting superiority over the aforementioned stochastic alternatives. In this work, we assess $\alpha$-entmax attention as baseline in §4. While both MapSelect and SPECTRA incorporate SparseMAP, MapSelect is specifically tailored for graph structures, allowing for both local and global applications.

## 6 Conclusion

We presented MapSelect, a method to learn sparse and interpretable attention scores in graph neural networks. MapSelect relies on SparseMAP, conventionally used in NLP (Guerreiro & Martins, 2021), to prune the attention scores both in a locally and globally controlled manner. The local approach, MapSelect-L, is more beneficial when we deal with node-centric tasks and want to enhance the sparsity at the surroundings of each node. The global approach, MapSelect-G, is more beneficial when we deal with graph-centric tasks and focus on the whole graph sparsity without any local constraints. Upon studying different trade-offs between sparsity, task performance, and interpretability, MapSelect-L achieved consistently the best performance w.r.t. different state-of-the-art alternatives in five datasets. Instead, MapSelect-G showed that it is more appropriate than alternative sparse solutions on denser graphs, where its stronger ability to control sparsity proved beneficial. By controlling the sparsity of the graph, the proposed approaches carry the potential advantage of overcoming over-smoothing Rathee et al. (2021) and over-squashing (Alon & Yahav, 2021). Such a task could be achieved by introducing different constraints into the MapSelect such as maximum spanning tree constraints and will be studied in future work. Furthermore, we note that MapSelect's flexibility opens the door for future research to easily incorporate it into various GNN architectures.

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

## A  $\alpha$-ENTMAX

The $\alpha$-entmax transformation (Peters et al., 2019) is a natural way to obtain a sparse attention distribution from a given vector of scores, $\boldsymbol{z} \in \mathbb{R}^n$. It is defined as the regularized argmax problem:

$$\alpha\text{-entmax}(\boldsymbol{z}) := \arg\max_{\boldsymbol{\pi} \in \triangle_{n-1}} \boldsymbol{z}^\top \boldsymbol{\pi} + H_\alpha(\boldsymbol{\pi}), \tag{9}$$

where $H_\alpha$ is a generalization of the Shannon and Gini entropies proposed by (Tsallis, 1988), parameterized by a scalar $\alpha \geq 0$:

$$H_\alpha(\boldsymbol{\pi}) := \begin{cases} \frac{1}{\alpha(\alpha-1)} \sum_j (\pi_j - \pi_j^\alpha), & \alpha \neq 1 \\ -\sum_j \pi_j \log \pi_j, & \alpha = 1. \end{cases} \tag{10}$$

Given the attention scores $\boldsymbol{z}_i \in \mathbb{R}^n$ of node $i$, the attention weights of $\alpha$-entmax can be computed in a thresholded form:[7]

$$\pi_{ij} = \alpha\text{-entmax}(\boldsymbol{z}_i)_j = [(\alpha - 1)z_{ij} - \tau(\boldsymbol{z}_i)]_+^{1/\alpha-1}, \tag{11}$$

where $[\cdot]_+$ is the ReLU function, and $\tau : \mathbb{R}^n \to \mathbb{R}$ is a normalizing function to ensure $\sum_j \pi_{ij} = 1$. Scalar $\alpha$ determines the propensity of sparsity: with $\alpha = 1$, $\alpha$-entmax simplifies to the softmax function, whereas for $\alpha > 1$, it returns sparse solutions. As $\alpha$ increases, the resulting probability distribution becomes more sparse. For $\alpha = 2$, the transformation recovers sparsemax (Martins & Astudillo, 2016), defined as the Euclidean projection of $\boldsymbol{z}_i$ onto the probability simplex. We refer to Peters et al. (2019) on how to compute $\tau(\cdot)$ efficiently in $O(n \log n)$.

We use $\alpha$-entmax in §4 as baseline. Remarkably, in a GAT setup, edges with a score $z_{ij} \leq \tau(\boldsymbol{z}_i)/\alpha-1$ will receive zero probability (i.e., $\pi_{ij} = 0$), and therefore can be excluded from the computation graph. Since $\alpha$-entmax promotes solutions that hit the boundary of the simplex (discouraging uniform distributions), it can mitigate the lack of expressiveness present in softmax-based GATs (Brody et al., 2022; Fountoulakis et al., 2023). Still, $\alpha$-entmax is restricted to produce solutions in the probability simplex, which may limit its applicability towards sparsifying the computation graph globally.

## B  EXPERIMENTAL SETUP

### B.1  TRAINING

For all models, we employ the cross-entropy loss for training and optimize the loss with Adam (Kingma & Ba, 2015). For MapSelect-L, we found that feeding SparseMAP with attention weights rather than raw attention scores works better in practice. Similarly, for MapSelect-G, we found that applying an exponential operation before passing scores to SparseMAP improves stability. We report average numbers of five distinct random seeds. We used a single machine equipped with a GeForce RTX 2080 Ti (11GB) GPU. We summarize relevant training hyperparameters in Table 2.

Table 2: Training hyperparameters.

| Hyperparam. | CiteSeer | Cora | PubMed | Actor | Amazon Photos | BA-Shapes |
|---|---|---|---|---|---|---|
| Hidden size | 8 | 8 | 8 | 8 | 8 | 20 |
| Dropout | 0.6 | 0.6 | 0.6 | 0.3 | 0.6 | 0.1 |
| Learning rate | 0.01 | 0.01 | 0.01 | 0.01 | 0.01 | 0.01 |
| Weight decay | 0.0005 | 0.0005 | 0.01 | 0.0005 | 0.0005 | 0.001 |

Concerning the model architecture, all approaches conduct their classification using two GAT layers. Methods that incorporate a masking layer, such as MapSelect and NeuralSparse, include an additional layer dedicated to learning a graph mask directly from the input. To ensure a balanced comparison across all methods, this masking layer does not modify the input features for the two classification layers, except for adjustments related to the graph itself. In the case of BA-Shapes,

---

[7]We drop the dependence on the layer $\ell$ for ease of exposition.

all models employ a standard GNN layer to encode all input features.The necessity of this initial pass arises from the fact that the standard GAT implementation alone is incapable of exclusively detecting the graph structure. For instance, in cases where all node feature vectors consist solely of '1', our GAT implementation will aggregate and normalize the surrounding feature vectors, yielding once again a feature vector of '1' for all nodes.

We present the hyperparameters used for controlling the sparsity of all methods employed in this work in Table 3. For SGAT, we set $\gamma$ to different values depending on the dataset. Specifically, we set $\gamma = 10^{-5}$ for CiteSeer, Cora and BA-Shapes, $\gamma = 10^{-6}$ for PubMed and Amazon Photo, and $\gamma = 10^{-7}$ for Actor.

Table 3: Configuration of hyperparameters used for controlling sparsity.

| Method | Hyperparam. | Values |
|---|---|---|
| SGAT | weight of $\ell_0$ penalty | $\{0, 1.0, 1.5, 2.2, 2.4, 2.6, 2.8, 3.0, 3.2, 3.4, 3.6, 3.8, 4.0, 5.0, 6.0\} \times \gamma$ |
| DropEdge | portion of dropped edges | $\{0.1, 0.2, 0.3, 0.4, 0.5, 0.6, 0.7, 0.8, 0.9, 1.0\}$ |
| NeuralSparse | maximum number of edges per node | $\{1, 2, 4, 6, 8, 10, 12, 16, 20, 25, 50, 100\}$ |
| Entmax | propensity to sparsity ($\alpha$) | $\{1.0, 1.5, 2.0, 2.5, 3.0, 3.5, 4.0, 4.5, 5.0, 6.0, 10.0\}$ |
| Top-$k$ | maximum number of edges per nodes | $\{1, 2, 4, 6, 8, 10, 12, 16, 20, 25, 50, 100\}$ |
| MapSelect-L | SparseMAP absolute budget ($B$) | $\{1, 2, 4, 6, 8, 10, 12, 16, 20, 25, 50, 100\}$ |
| MapSelect-G | SparseMAP percentage budget ($B$) | $\{0.1, 0.2, 0.3, 0.4, 0.5, 0.6, 0.7, 0.8, 0.9, 1.0\}$ |

## B.2 METRICS

We evaluate interpretability with the sparsity and fidelity metrics proposed by BAGEL (Rathee et al., 2022) and ZORRO (Funke et al., 2023), defined next. These scores are calculated over the explanations of 300 randomly selected nodes. The random selections are kept consistent for each dataset. Both metrics are evaluated against the percentage of *removed edges*, where an edge is considered removed when its explanation score is zero.

**Rationale sparsity.** Computes the Shannon entropy over the explanation vector $\boldsymbol{p} \in \triangle_{n-1}$:

$$H(\boldsymbol{p}) = -\sum_i p_i \log p_i. \tag{12}$$

**RDT-fidelity.** Given explanations $\boldsymbol{p}_i \in \triangle_{n-1}$ for each node $1 \le i \le n$, let $\boldsymbol{M} \in [0, 1]^{n \times n}$ denote a mask matrix, such that $M_{ij} = p_{ij}$. The RDT-Fidelity concerning the network $\Phi$ and the noise distribution $\mathcal{N}$, is expressed as follows:

$$\mathcal{F}(\boldsymbol{M}) = \mathbb{E}\left[1_{\Phi(\boldsymbol{X}) = \Phi(\tilde{\boldsymbol{X}}(\boldsymbol{M}))}\right], \tag{13}$$

where $\boldsymbol{X} \in \mathbb{R}^{n \times n}$ represents the input, and $\tilde{\boldsymbol{X}}(\boldsymbol{M})$ is a perturbed input defined as:

$$\tilde{\boldsymbol{X}}(\boldsymbol{M}) = \boldsymbol{M} \odot \boldsymbol{X} + (1 - \boldsymbol{M}) \odot \boldsymbol{Z}, \quad \boldsymbol{Z} \sim \mathcal{N}. \tag{14}$$

As Rathee et al. (2022), we set the noise distribution as the global empirical distribution of the input features.

## B.3 DATASETS

As a node classification task the Cora, PubMed, CiteSeer, Actor and Amazon Photo datasets (Yang et al., 2016; Pei et al., 2020; Shchur et al., 2018) are evaluated using the default configurations provided by PyTorch Geometric (Fey & Lenssen, 2019). These datasets can be classified as *transductive*, indicating that there is no isolation of the training set from the validation set as all data points are part of a single graph. We provide an overview of the datasets in Table 4.

To evaluate our method with ground-truth explanations, we opted for the Barabasi-Albert (BA-Shapes) dataset (Ying et al., 2019). This is a dataset with 300 random nodes and a set of 80 "house"-structured graphs connected to it. The dataset contains 4 classes; a node can be classified as the top, the middle or the bottom of a house or as not being part of a house. Each node has a single input feature equal to 1, *forcing the network to only classify based on the graph structure*. As a ground truth, an edge and node mask are passed containing all edges and nodes that are part of a house-structure.

Table 4: Overview of the datasets used in our experiments.

|  | # nodes | # edges | # features | # classes |
|---|---|---|---|---|
| **Cora** | 2,708 | 10,556 | 1,433 | 7 |
| **CiteSeer** | 3,327 | 9,104 | 3,703 | 6 |
| **PubMed** | 19,717 | 88,648 | 500 | 3 |
| **Actor** | 7,600 | 30,019 | 932 | 3 |
| **Amazon Photo** | 7,650 | 238,162 | 500 | 3 |
| **BA- Shapes** | 700 | 3936 | 1 | 4 |

## C EXPLANATION EXTRACTION

**Node-level explanation.** Node-level explanations are generated by (i) setting the value of edge weights as the attention weights of GAT network that produced the original classification, and (ii) propagating an identity matrix of size $N$, the number of nodes in the graph. We can formally describe this as follows. Let $\tilde{\pi}_{ij}^{(\ell)} \in \mathbb{R}$ be the attention weight associated with the edge between node $i$ and its neighbour $j$ at layer $\ell$ in the original GAT network, extracted after applying an interpretability method. For example, in MapSelect, $\tilde{\pi}_{ij}^{(\ell)}$ is masked according to SparseMAP's output and then re-normalized, as stated in the left part of Equation 6. Overall, for each explainability method, we perform the following steps to extract node-level explanations:

1. Recover the attention weights $\tilde{\pi}_{ij}^{(\ell)}$ by running the GAT network on the input graph with original feature vectors $\boldsymbol{h}_i \in \mathbb{R}^d$, for each node $i$.

2. Create a new one-hot vector representation for node $i$, $\boldsymbol{h}_i^{(0)} = \{0, 1\}^N$, where $h_{ij} = 1$ if $i = j$ and $h_{ij} = 0$ otherwise.

3. Propagate the new representation through a weighted-message passing network, with as many layers as the original network. That is, we compute new node features as follows:

$$\boldsymbol{h}_i^{(\ell+1)} = \sum_{j \in \mathcal{N}_i} \tilde{\pi}_{ij}^{(\ell)} \boldsymbol{h}_j^{(\ell)}, \tag{15}$$

where $\mathcal{N}_i$ represents the set of neighbors of node $i$. That is, the new node representation is simply a weighted sum of one-hot vectors.

4. Obtain the explanation for node $i$ from its final node features ($\boldsymbol{h}_i^{(\text{final})} \in \mathbb{R}^N$):

$$\boldsymbol{p}_i = \frac{\boldsymbol{h}_i^{(\text{final})}}{\sum_{j=1}^N \boldsymbol{h}_{ij}^{(\text{final})}} \in \triangle_{N-1}, \tag{16}$$

where $p_{ij}$ represents the importance of node $j$ to the classification of node $i$. Therefore, to get a final node-level explanation with respect to a target node $i^\star$, we simply extract $\boldsymbol{p}_{i^\star}$.

Note that all nodes outside of the computation graph of node $i$ will receive an importance score of zero. When calculating the fidelity and sparsity scores, these importance scores are not included.

**Calculating fidelity and sparsity.** The fidelity and sparsity scores are calculated over each extracted node-level explanation. The scores have been computed and averaged for 300 randomly selected nodes. For each dataset, the same 300 nodes were used to evaluate all methods.

## D ADDITIONAL RESULTS

### D.1 REAL-WORLD DATASETS

**Sparsity-entropy tradeoff.** Figure 6 shows the impact of sparsification on entropy (described in §B.2), where a low entropy indicates a more focused rationale. The initial performance of Entmax and top-$k$ can be attributed to a better allocation of the attention distribution. This is seen by the lower entropy of Entmax, indicating a more focused explanation.

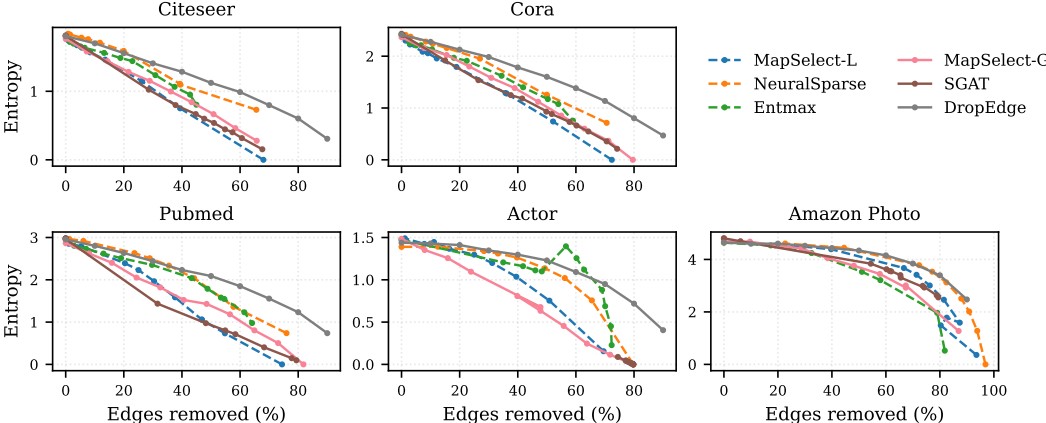

Figure 6: Trade-off between graph sparsity and explanation sparsity.

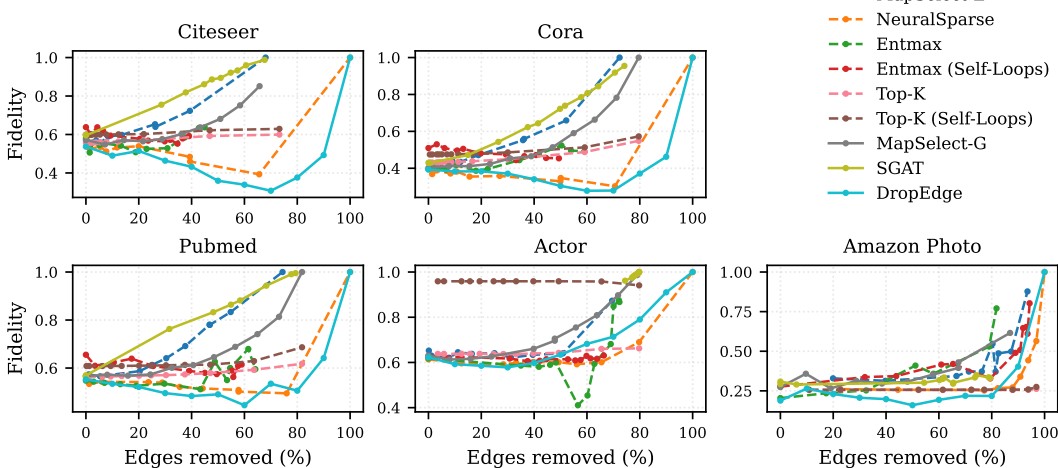

Figure 7: Trade-off between fidelity and sparsity, with and without maintaining self-loops.

**Sparisty-interpretability tradeoff.** Forcing the model to maintain the self-loops (not allowing them to be masked out) greatly improved the fidelity scores as shown in Figure 7. Only Entmax produces a better result when not maintaining the self-loops, however, this approach also presented more instability. As a remark, in the main paper we provide the Entmax version that preserves the self-loops for the sake of consistency.

## D.2 Synthetic dataset

**Tradeoffs.** In Figure 8a, we illustrate the trade-off between sparsity and performance in the BA-Shapes dataset. Interestingly, even with the removal of all edges, an accuracy of 85% is achieved. This can be attributed to the initial pass through a single GNN layer for all methods, since this layer helps all methods to learn sparse subgraphs. In addition to assessing the AUC score, we also conducted an evaluation of the extracted rationales in relation to the ground truth in terms of raw accuracy, as depicted in Figure 8b. For this, we set a threshold of 0.5 for binarizing explanations. Intuitively, the accuracy metric applies a greater penalty to values approaching zero rather than exactly zero, treating them with the same severity as values that are higher but still fall below the threshold. This accounts for the disparity observed in Figure 4, where the AUC score exhibits differing starting points due to certain models learning weights that approach zero more than other models. Via this accuracy score, the trend of explanations improving as more edges are removed becomes even more pronounced.

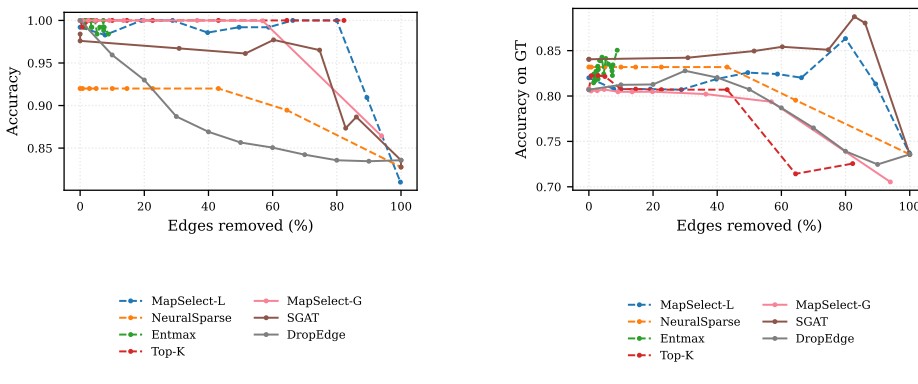

(a) Trade-off between sparsity and task accuracy.

(b) Trade-off between sparsity and interpretability (accuracy).

Figure 8: Additional results on the BA-Shapes dataset.

**Explanation example.** In the BA-Shapes dataset, edges that do not pertain to a "house" structure are considered irrelevant. We anticipate that MapSelect will effectively filter out these non-structural edges, offering the remaining edges as a rationale. As illustrated in Figure 9, a subgraph from the BA-Shapes dataset showcases the attention weights learned in one of our experiments. As anticipated, the majority of attention is directed towards the edges constituting this house-like structure.

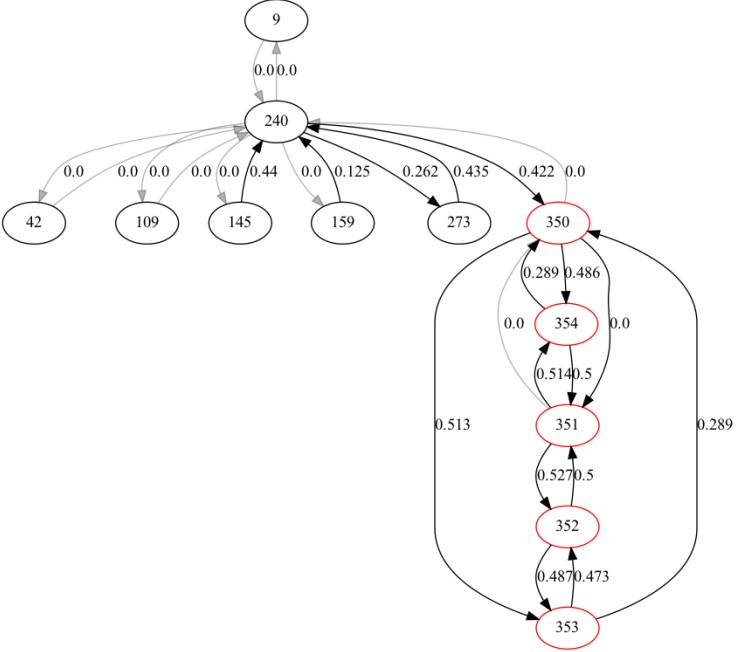

Figure 9: Example of the generated attention values by *MapSelect-L* on the BA-Shapes dataset with a budget of $B = 2$. Here we show all nodes within a $k$-hop distance of 2 from node 350. A red border indicates that a node is part of a "house".

### D.3 CONTROL OVER SPARSITY

Here, we investigate the sensitivity of the sparse hyperparameters in SGAT and MapSelect-G towards the actual computed sparsity levels (percentage of edges removed from the graph), employing two dense datasets, namely, CiteSeer and PubMed. Results are shown in Figures 10 and 11.

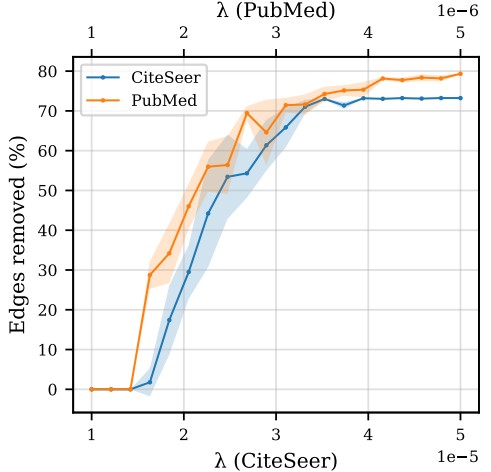

Figure 10: Amount of edges removed according to the sparsity parameter $\lambda$ in SGAT.

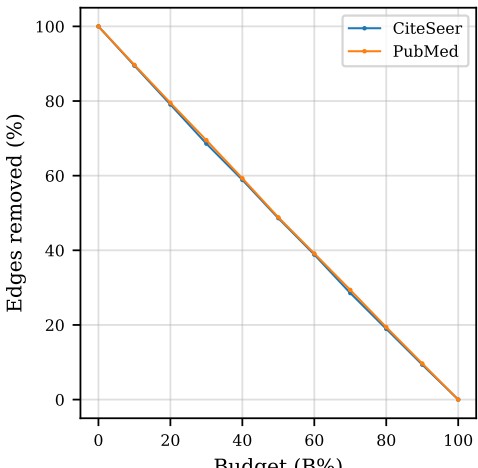

Figure 11: Amount of edges removed and the sparsity budget $B\%$ in MapSelect-G.

