# OpenReview forum: "MapSelect: Sparse & Interpretable Graph Attention Networks"
_ICLR.cc/2024/Conference — Submitted to ICLR 2024_

### Official Review · Reviewer_b971 · 2023-10-27

**Soundness:** 2 fair
**Presentation:** 2 fair
**Contribution:** 2 fair
**Rating:** 3
**Confidence:** 4

**Summary:**

In this article, the authors explore the limitations of Graph Attention Networks (GATs), particularly concerning their dense attention weights, which allocate probability mass to all neighbors of a node. This denseness can dilute the model's focus, as every edge is given some attention. To address this issue, the authors introduced a novel, fully differentiable sparse attention mechanism that allows for precise control over attention density. This mechanism is a continuous relaxation of the conventional top-k operator, facilitated through user-defined constraints.

The authors propose two unique variations of this sparse attention mechanism:
1. A local approach that ensures a consistent degree per node.
2. A global method that retains a specific percentage of the complete graph.

Through a comprehensive evaluation, they assess the five sparse GATs on metrics like sparsity, performance, and interpretability, offering valuable insights into the trade-offs between accuracy and both sparsity and interpretability. The findings reveal that the new sparse attention mechanism offers superior representability over baseline models, with the local approach showing promise.

**Strengths:**

Originality: Fair. This work combines two existing ideas, GAT and SparseMap. This work is a minor variation of a well-studied problem.

Quality: Fair. In the experiment part, this work compared two variants MapSelect-L and MapSelect-G with several baselines such as Top-k, Entmax, NerualSpare, SGAT, DropEdge, which is a good practice in empirical evaluations.

Clarity: This work provides a lucid explanation of the proposed methodology. The appendix also contains a wealth of details.

Significance: The provided method outperforms some baselines in interpretability in some cases.

**Weaknesses:**

Novelty: The problem of improving differentiability by integrating some mechanisms is not unique to GATs. Mathematical novelty might seem limited. Looking forward to seeing the improvements in theory.

Technical: The tool SparseMap to sparse the graph is an open-source one, but the code has not been updated for 5 years. Looking forward to seeing the update in the code.

Presentation: Using both American English and British English, such as "neighboring" and "neighbours" in the same article, can be confusing for readers and is generally not considered good practice. Consistency in spelling, grammar, and style within a single document helps maintain clarity and professionalism.

**Questions:**

In the paper, a novel sparse attention mechanism was introduced to address the limitations of dense attention in Graph Attention Networks (GATs). While the approach is intriguing, I noticed that the improvements over the state-of-the-art, were not considerably very significant, especially when considering the added complexity of your proposed mechanism. Can you elaborate on the tangible benefits of adopting this new method over existing approaches, especially in real-world applications where computational efficiency might be crucial? Are there specific scenarios or types of data where your method shows distinct advantages?

---

> ### Author Response · Authors · 2023-11-22
>
> Thank you for your thoughtful review and insightful observations regarding our manuscript. We appreciate the time you took to analyze our work and provide constructive feedback. Below, we address your concerns and questions:
>
> 1. **Novelty and Theoretical Contribution:** We understand your point about the mathematical novelty appearing limited. Our aim was to address a specific practical limitation in GATs by combining SparseMap with GATs in a novel way. However, we believe the paper, as is, offers valuable empirical analysis alongside a clear mathematical description of our method.
>
> 2. **Code and Tool Update:** We appreciate your note on the usage of SparseMap and its code update status. We agree that maintaining and updating the tools we rely on is crucial. For this reason, we have documented the SparseMap part of the code, ensuring that it remains a viable tool in the field.
>
> 3. Consistency in Writing Style: Your observation regarding the inconsistent use of American and British English is well taken. We apologize for any confusion this may have caused and we have revised manuscript to maintain consistency in spelling and grammar.
>
> 4. **Benefits Over Existing Approaches:** We acknowledge that the improvements in terms of accuracy might not seem significantly higher than state-of-the-art methods. However, the primary advantage of our approach over SGAT, lies in its ability to offer precise control over attention density. This control facilitates a more interpretable and sparse representation (see answer to Reviewer EqGL).  On top of this, in contrast to SGAT, MapSelect also works locally, and from our evaluation, is the best performing local-based method. We believe these two points make a significant step forward in understanding how the model allocates attention.
>
> 5. **Applicability in Specific Scenarios:** MapSelect shows particular strengths in scenarios where interpretability is as important as performance, such as in domains where decisions need to be explainable, justifiable, and adhere to a certain format (e.g., at most three nodes/edges), like healthcare or finance.
>
> Thank you once again for your valuable feedback.

---

### Official Review · Reviewer_EqGL · 2023-10-30

**Soundness:** 2 fair
**Presentation:** 2 fair
**Contribution:** 1 poor
**Rating:** 3
**Confidence:** 3

**Summary:**

The paper introduces a novel method called MapSelect to address issues with dense attention layers in Graph Attention Networks (GATs). The proposed method offers a fully differentiable sparse attention mechanism that allows precise control over attention density, acting as a continuous relaxation of the top-k operator. The paper presents two versions of MapSelect: a local approach maintaining a fixed degree per node and a global approach preserving a percentage of the full graph. The approach seems to be effective but lacks rigorous analysis/proof. Finally, it tests on several widely-used datasets, however the experimental study seems not convincing.

**Strengths:**

1.	It tests on several widely-used datasets, and the proposed method can sometimes beat the existing methods.

**Weaknesses:**

1.	The core part (the proposed method in Section 3) lacks of sufficient analysis. We know that it is not difficult to put different modules together to form a paper. But we should make sure that the motivation of doing that really makes sense and we should understand what we are doing.
2.	The writing needs to be largely improved. The content in introduction is hard to follow. The symbol system needs to be improved.
3.	As shown in Figures 2-5, it seems that SGAT performs better than the proposed methods? As we can see that SGAT almost always stay on the highest line.

**Questions:**

1.	See the weakness in the “*Weaknesses” part.

---

> ### Author Response · Authors · 2023-11-22
>
> Thank you for your insightful review and constructive criticism of our manuscript. We value your feedback and agree that addressing these points will enhance the quality of our paper. Below, we address each of your concerns:
>
> 1. **Lack of Sufficient Analysis:** We have enhanced the clarity of our motivations in the introduction. Moreover, we maintain that our method is theoretically robust, as substantiated in Sections 2 and 3. Notably, unlike prior approaches, our method has the unique capability to explain both local and global contexts.
>
> 2. **Improvement in Writing and Symbol System:** We regret that our writing style and symbol system were not clear in the introduction and other sections of the paper. To address this, we will undertake a thorough review and revision of our manuscript, focusing on improving readability and clarity. We would greatly appreciate it if you could provide specific sentences or paragraphs where you found the content particularly challenging to follow. This will help us target our revisions more effectively.
>
> 3. **Performance Comparison with SGAT:** Your observation regarding SGAT's performance in comparison to our proposed methods is valid. However, it's important to emphasize that our method works both locally and globally, with superior performance on the local setting. Moreover, it offers significantly greater control over sparsity, which is a crucial aspect of interpretability in GATs. To clarify this advantage, we have included two additional plots in our revised manuscript (see Figures 10 and 11 in Appendix D.3). These plots showcase the relationship between the hyperparameters controlling sparsity (e.g., l1 penalty for SGAT and budget for MapSelect) and the resultant sparsity levels, thus highlighting the controllability and interpretability advantages of MapSelect over SGAT.
>
> We believe these revisions will address your concerns and significantly improve the manuscript. Thank you again for your valuable feedback, and we look forward to resubmitting our improved paper.

---

### Official Review · Reviewer_iHh7 · 2023-10-30

**Soundness:** 3 good
**Presentation:** 3 good
**Contribution:** 2 fair
**Rating:** 3
**Confidence:** 5

**Summary:**

This paper aims to improve the explainability of attention weights of graph attention networks. In particular, the paper proposes MapSelect, which adopts SparseMap to provide fully differentiable sparse attention mechanism for better explainability. MapSelect enables precise control over the attention density. The paper proposes two variants of MapSelect, a local approach maintaining a fixed degree per node and a global approach preserving a percentage of the full graph. Experimental results show accuracy-interpretability trade-off when sparsity increases.

**Strengths:**

1. The proposed method is simple and makes sense
2. The proposed method enables precise control over the attention density
3. Experimental results show that MapSelect can improve the interpretability of graph attention network

**Weaknesses:**

1. The accuracy and interpretability of the proposed method are just comparable with SGAT on most datasets used in the paper. In other words, it doesn’t significantly outperform the baselines.
2. From experimental results, we can find that the performance drops significantly when the graph becomes sparse. It is doubtable if one would like to sacrifice the accuracy. Thus, the paper should also add one baseline, i.e., adopts GAT as a baseline for accuracy and adopts a post-hoc explainer for GAT for explainability. If such simple baseline outperforms the proposed method in terms of both accuracy and interpretability, or has comparable interpretability and better accuracy, then the proposed method is unnecessary.
3. The scalability of the proposed MapSelect-G is questionable as one need to solve SparseMap problem in each epoch. It is better to provide time complexity analysis and real running time on the used datasets. In addition, the datasets used are very small. The authors should also conduct experiments on larger datasets.
4. Currently, the proposed method is only limited to graph attention network. The authors might consider replacing the second GAT in Figure 1(A) by other GNN models to show that the proposed method is flexible to benefit various GNNs

**Questions:**

Please see above.

---

> ### Author Response · Authors · 2023-11-22
>
> Thank you for your thoughtful and detailed review of our manuscript. We appreciate your insights and agree that addressing these points will strengthen our paper. Below, we respond to each of your concerns:
>
> 1. **Accuracy-Interpretability Trade-off:** We acknowledge your observation regarding the comparable performance of MapSelect with SGAT. It is indeed a characteristic of self-interpretable methods to potentially trade off accuracy for increased sparsity and interpretability. We realize this might not have been sufficiently emphasized in our paper. In our revised manuscript, we have clarified this trade-off in the introduction, highlighting that our primary goal is to improve interpretability, even if it sometimes comes at the expense of accuracy.
>
> 2. **Baseline Selection:** Regarding your suggestion to include a baseline combining GAT with a post-hoc explainer, we understand the rationale behind it. However, our work intentionally focuses on self-interpretable models. While the proposed baseline is valuable, it falls outside our paper's scope, which is dedicated to exploring intrinsic interpretability within GNNs rather than relying on external explainers. We have made our scope and the rationale behind our choice of baselines clearer in the revised paper.
>
>
> 3. **Scalability Concerns:** SparseMAP enjoys a convex optimization problem and has a very efficient implementation in C++. Its convergence can also be controlled via hyperparameters. In practice, we found that MapSelect is 2x (local) and 4x (global) faster than NeuralSparse. This shows that runtime performance is not a bottleneck.
>
> 4. **Flexibility with Various GNN Models:** Your suggestion to demonstrate the flexibility of our method by integrating it with different GNN models is intriguing. While our current focus is on graph attention networks, we acknowledge the potential value in showing the method's applicability to other GNN architectures. However, integrating MapSelect with non-attention-based GNNs might change the nature of the explanations provided, as our method is designed to interpret GATs. We will consider exploring this in future work and we have noted this as a potential benefit of our approach.
>
> Thank you once again for your valuable feedback.

---

### Official Review · Reviewer_WU6w · 2023-11-01

**Soundness:** 2 fair
**Presentation:** 4 excellent
**Contribution:** 2 fair
**Rating:** 3
**Confidence:** 4

**Summary:**

Graph Attention Networks (GATs) can capture complex graph structures, but the dense nature of softmax functions gives non-zero probability to even irrelevant neighbors. To tackle this problem, this paper proposes MapSelect, a sparse attention mechanism as an alternative to GATs. MapSelect allows to control of the attention sparsity using a continuous relaxation of the top-k operator. The authors demonstrate the sparsity, performance, and interpretability of MapSelect on six benchmarks including one synthetic dataset.

**Strengths:**

This paper tackles an important problem of graph representation learning, the sparsity, and the interpretability of graph attention networks. The proposed method is easy and straightforward, and effective in terms of performance and interpretability.

**Weaknesses:**

- The stated strengths (Section 2: deterministic and end-to-end differentiable, and thus easier to optimize) are not fully justified. For the ‘deterministic, thus easier to optimize’ part, is it really true? Is there any reference? For the ‘end-to-end differentiable, thus easier to optimize’ part, why do authors think that existing works (NeuralSparse and SGAT) are not end-to-end differentiable? The reparameterization makes their methods end-to-end differentiable.
- This paper compares MapSelect with sparse GAT variants. However, a comparison with the original GAT is needed in terms of performance and efficiency. Since the sparse-masking procedure requires additional computations and memory, there is a need to accurately quantify the benefits gained over the original GAT.
- The interpretability is not deeply investigated. Only one dataset (BA-Shapes) for graph-level explanation tasks is used. There is no comparison with other GNN explainers other than attention. In addition to fidelity, label-agreement for node-level tasks can be a good metric for attention quality (from How to Find Your Friendly Neighborhood: Graph Attention Design with Self-Supervision, ICLR 2021)
- Masking attention for sparsity is also studied in Transformers. Two missing papers below are about sparse Transformers for graph-structured data.
  - Transformers meet Stochastic Block Models: Attention with Data-Adaptive Sparsity and Cost, NeurIPS 2022
  - EXPHORMER: Sparse Transformers for Graphs, ICML 2023
- There are some missing results:
  - Top-k attention is missing in Figure 2.
  - Where can I find the results stated in “both MapSelect-G and SGAT outperform the AUC scores presented by the attention, gradient, and GNNExplainer baselines in Luo et al. (2020), with SGAT even outperforming the proposed PGExplainer itself.”
- Typos: EWe (Section 3) → We

**Questions:**

- Section 2.1 uses three times of parameters against the original GAT implementation: W1 (d’ x 2d) + W2 (d’ x d) = 3 *  W (d’ x d) in GAT. Is this modification applied to not only MapSelect and other baselines? We can validate this if the authors upload their codes in the supplementary material.
- The term ‘computation graph’ can be misunderstood as 'graphs where nodes are mathematical operations'. What about input graphs?

---

> ### Author Response · Authors · 2023-11-22
>
> Thank you for your thorough review and constructive feedback on our paper. We appreciate the opportunity to clarify and enhance our work based on your insights. Below, we address each of your concerns:
>
> 1. **End-to-End Differentiability and Determinism:** We acknowledge that our claim regarding determinism and end-to-end differentiability was not fully substantiated in the initial manuscript. We agree that the reparameterization trick, as used in NeuralSparse and SGAT, renders these methods end-to-end differentiable. To clarify, our method's deterministic nature was intended as a strength, following the comparison in SPECTRA [(Guerreiro & Martins, 2021)](https://arxiv.org/abs/2109.04552) between deterministic and stochastic approaches, including the reparameterization trick. We have added the necessary citation to SPECTRA to support this claim.
>
>
> 2. **Comparison with Original GAT:** We recognize the importance of comparing MapSelect with the original GAT in terms of performance and efficiency. While MapSelect does incorporate SparseMAP, leading to slower performance compared to standard GATs, our primary focus is not on performance gains over the original GAT. Instead, we emphasize the interpretability and sparsity benefits. You can find the result of a standard GAT (i.e., without sparsity) when looking at the first point of Entmax (alpha = 1.0) and MapSelect (budget = 100%).  However, to provide a comprehensive view, we have made this distinction clearer in our revised manuscript (see footnote 6).
>
> 3. **Focus on Self-Interpretable Models in Node-Classification Contexts:** We appreciate your suggestion regarding the expansion of our interpretability analysis. It's important to highlight that our main goal in this paper was to investigate self-interpretable models specifically in the context of node-classification datasets. This angle of analysis, focusing on self-interpretability in node-classification, represents a novel contribution to the field, as far as we are aware. We agree that incorporating both node-level and graph-level classification tasks will enrich our study. Do you have recommendations for graph-level classification datasets?
>
> 4. **Omission of Top-k Attention in Figure 2:** The exclusion of top-k attention from Figure 2 was intentional, as its results are similar to those of normal GAT (we only apply top-k at test time, see Sec. 4.1).
>
> 5. **Comparison with the results in Luo et al. (2020):** We have provided the specific section reference in Luo et al. (2020) where the results we mentioned can be found, for the convenience of readers.
>
> 6. **Clarification on Parameter Use in Section 2.1:** Thank you for your observation regarding the parameter usage in our model. We realize there might be a misunderstanding due to the conventional notation used in the field. In our implementation, contrary to what might be inferred from the notation, we actually utilize two distinct matrices (W1 and W2), and not a single matrix as is often represented (W) in other papers. However, note that these two matrices can be seen as a single W applied after the concat operation.
>
> Thank you again for your valuable feedback. We believe these revisions significantly strengthed our paper.

---

> > ### Comment · Reviewer_WU6w · 2023-11-23
> >
> > Thank you for your response.
> >
> > ---
> >
> > ```
> > While MapSelect does incorporate SparseMAP, leading to slower performance compared to standard GATs, our primary focus is not on performance gains over the original GAT. Instead, we emphasize the interpretability and sparsity benefits.
> > ```
> > Sparsity is usually related to efficiency in the machine learning field, and potential readers can be confused about this. I would recommend that authors focus on interpretability. Plus, even though the authors' focus is not on efficiency, the readers might want to know the trade-off between interpretability and efficiency.
> >
> > ---
> >
> > ```
> > We agree that incorporating both node-level and graph-level classification tasks will enrich our study. Do you have recommendations for graph-level classification datasets?
> > ```
> >
> > This link is related to your question: https://pytorch-geometric.readthedocs.io/en/latest/modules/datasets.html#synthetic-datasets

---

### Meta-Review · Area_Chair_9gyh · 2023-11-29

**Metareview:**

The paper addresses limitations of Graph Attention Networks (GATs) due to their dense attention weights, which assign probability to all node neighbors, potentially diluting focus. To mitigate this, the authors introduce MapSelect, a fully differentiable sparse attention mechanism offering precise control over attention density via a continuous relaxation of the top-k operator. MapSelect includes local and global variants, ensuring a fixed degree per node or preserving a percentage of the full graph, respectively. The study evaluates five sparse GATs on sparsity, performance, and interpretability metrics, highlighting an accuracy-interpretability trade-off with increasing sparsity. Results demonstrate MapSelect's effectiveness in enhancing model representability, particularly the local approach, though the paper lacks rigorous analysis. The experimental study, while conducted on widely-used datasets, may require further validation for robust conclusions.

The suggested approach is both straightforward and impactful. However, the paper could benefit from a more thorough discussion on various differentiable sparse attention methods currently available. Therefore, I recommend that the authors revise the paper to address the comments from the reviewer and resubmit it to a future venue.

**Justification For Why Not Higher Score:**

Since all the reviewers vote for rejection and I agree with their assessment, I also vote for rejection.

**Justification For Why Not Lower Score:**

N/A

---

### Decision · Program_Chairs · 2024-01-16

Reject